# Glucose Favors Lipid Anabolic Metabolism in the Invasive Breast Cancer Cell Line MDA-MB-231

**DOI:** 10.3390/biology9010016

**Published:** 2020-01-10

**Authors:** Mª Carmen Ocaña, Beatriz Martínez-Poveda, Ana R. Quesada, Miguel Ángel Medina

**Affiliations:** 1Departamento de Biología Molecular y Bioquímica, Facultad de Ciencias, Andalucía Tech, Universidad de Málaga, E-29071 Málaga, Spain; mc.ocanaf@gmail.com (M.C.O.); bmpoveda@uma.es (B.M.-P.); quesada@uma.es (A.R.Q.); 2IBIMA (Biomedical Research Institute of Málaga), E-29071 Málaga, Spain; 3CIBER de Enfermedades Raras (CIBERER), E-29071 Málaga, Spain

**Keywords:** cancer, lipid metabolism, glucose metabolism, glutamine metabolism

## Abstract

Metabolic reprogramming in tumor cells is considered one of the hallmarks of cancer. Many studies have been carried out in order to elucidate the effects of tumor cell metabolism on invasion and tumor progression. However, little is known about the immediate substrate preference in tumor cells. In this work, we wanted to study this short-time preference using the highly invasive, hormone independent breast cancer cell line MDA-MB-231. By means of Seahorse and uptake experiments, our results point to a preference for glucose. However, although both glucose and glutamine are required for tumor cell proliferation, MDA-MB-231 cells can survive two days in the absence of glucose, but not in the absence of glutamine. On the other hand, the presence of glucose increased palmitate uptake in this cell line, which accumulates in the cytosol instead of going to the plasma membrane. In order to exert this effect, glucose needs to be converted to glycerol-3 phosphate, leading to palmitate metabolism through lipid synthesis, most likely to the synthesis of triacylglycerides. The effect of glucose on the palmitate uptake was also found in other triple-negative, invasive breast cancer cell lines, but not in the non-invasive ones. The results presented in this work suggest an important and specific role of glucose in lipid biosynthesis in triple-negative breast cancer.

## 1. Introduction

Tumor metabolism has been studied since the early 20th century. In 1925, Otto Warburg observed that tumor cells produced greater amounts of lactic acid than other healthy tissues, even in the presence of oxygen [1]. Later, it would be known that these cells are able to obtain the same amount of energy by glycolysis than by oxidative phosphorylation (OXPHOS). This fact would afterwards be known as the Warburg effect [2]. Since the beginning of the present century, the metabolism of tumor cells has regained great importance, and metabolic shift was added as a new hallmark of cancer [3]. As a result, tumor metabolism has been widely studied in the last years, and multiple reviews have been published focusing on the Warburg effect, glutamine utilization, and the genetic alterations involved in these processes [4,5,6,7].

Breast cancer is the second leading cause of cancer death among women [8]. Nevertheless, mortality from breast cancer in North America and the European Union has decreased because of early detection and the development of efficient therapies, thanks to the increasing knowledge on the processes occurring inside tumor cells. Because of the importance of the metabolic reprogramming in cancer, the metabolism of breast cancer cells has been exhaustively studied in the last years in order to know more about this disease and to design more effective therapies.

MDA-MB-231 is one of the most studied estrogen receptor (ER) negative breast cancer cell lines. This cell line is known to be highly metastatic, invasive, and glycolytic [9]. However, most of the metabolic studies are usually performed after incubations of no less than 24 h, often because they are focused on the effects on the proliferation rate and the invasive capacity of these cells. Moreover, palmitate is one of the most abundant fatty acids (FAs), but it has been shown to induce apoptosis in MDA-MB-231 and other breast tumor cell lines [10]. Nevertheless, lipid metabolism is important for MDA-MB-231 tumorigenesis [11]. In this work, we wanted to study two features of MDA-MB-231 cells metabolism, namely: (1) the effect of glucose and/or glutamine starvation on proliferation, and (2) the possible alterations in metabolism after a short-time incubation with different combinations of glucose, glutamine, and/or palmitate. In order to avoid possible interferences with other metabolites from culture media and serum, in this work, we performed the short-time experiments in very restrictive conditions, although it should be taken into account that other metabolites may also be present in an individual. Thus, this kind of methodological approximation allows for the determination of the short-time preference of a metabolic substrate in these cells when only these fuels are present at physiological concentration after a fasted period [12,13]. Combining long- and short-term experiments covers different possible scenarios, such as the short-term response of the cells to a drastic nutritional change (their immediate fuel preference) and the long-term consequence of that change (their ability to adapt to a different nutritional condition). Increasing knowledge in the metabolic flexibility of cancer cells could allow for the design of specific therapies in order to inhibit tumor progression. Therefore, our study may lead to further research in the metabolomics of breast cancer cells and the inhibition of breast cancer progression.

The results obtained point to a preference for glucose, but also a strong dependence on both glucose and glutamine to proliferate. Furthermore, glucose increased the palmitate uptake selectively in MDA-MB-231 and other invasive breast cancer cell lines, but not in the non-invasive breast cancer cell line MCF7. This glucose would be most likely converted to glycerol-3-phosphate (G3P) in order to exert this effect, leading to lipid synthesis from palmitate.

## 2. Materials and Methods

### 2.1. Materials

Cell media, antibiotics, and trypsin were obtained from BioWhittaker (Verviers, Belgium). Fetal calf/bovine serum (FBS) was supplied by Biowest (Kansas, USA). Molecular probes (Eugene, OR, USA) provided us with BODIPY (BOron-DIPYrromethene) FL C_16_ and 2-NBDG. L-[^14^C(U)]-glutamine was supplied by Perkin Elmer (Waltham, MA, USA). Abnova (Taoyuan City, Taiwan) was the provider of the lactate assay kit. The antibodies used in the present study were purchased from Cell Signaling Technology (Danvers, MA, USA). Adipostatin-A and PD98059 were obtained from Cayman Chemical (Ann Arbor, MI, USA). The plastic material for the cell culture was from Nunc (Roskilde, Denmark). All of the other reagents were from Sigma-Aldrich (St. Louis, MO, USA). The palmitate-BSA conjugate was prepared as previously described [14].

### 2.2. Cell Culture

Unless specified otherwise, the cell culture media included glutamine (2 mM), 10% fetal bovine serum (FBS); and the antibiotics penicillin (50 U/mL), streptomycin (50 U/mL) and amphotericin (1.25 μg/mL). The tumor cells used in this paper were provided by the ATCC (Rockville, MD, USA) or the European Collection of Cell Cultures. Human breast carcinoma MDA-MB-231, MDA-MB-436, and HCC1937 cell lines and human neuroblastoma Kelly cells were maintained in RPMI-1640. Human breast carcinoma MCF7 cells were maintained in Dulbecco′s modified Eagle′s medium (DMEM) containing glucose (4.5 g/L). The human cervix adenocarcinoma HeLa cells were maintained in Eagle’s minimum essential medium (EMEM). Fresh human umbilical vein endothelial cells (HUVEC) were obtained as previously described [15], and maintained in a 199 medium supplemented with 20% FBS, endothelial cell growth supplement (ECGS) (30 µg/mL), and heparin (100 µg/mL). All of the cell lines were maintained at 37 °C under a humidified 5% CO_2_ atmosphere.

### 2.3. Cell Growth Curves

The MDA-MB-231 and MCF7 cells were seeded at a density of 1.5 × 10^4^ cells in 24-well plates. After cell adherence, the cells were incubated in normoxia or hypoxia (1% O_2_) in the presence or absence of glucose and/or glutamine. A Coulter counter from Beckman Coulter (Brea, CA, USA) was used to count the cell number every day.

### 2.4. Cytotoxicity Assays

The cells were seeded at a density of 2 × 10^4^ cells in 96-well plates, and 24 h after seeding were treated with or without 0.5 mM palmitate for an additional 16 h. The MTT (3-(4,5-dimethylthiazol-2-yl)-2,5-diphenyltetrazolium bromide) assay was carried out as described by us elsewhere [16].

### 2.5. Extracellular Flux Analyzer Experiments

MDA-MB-231 were cultured at a density of 5 × 10^4^ cells/well in 24-well Seahorse XF^e^24 plates (Agilent). The cell treatment and oxygen consumption rate (OCR)/extracellular acidification rate (ECAR) measurements were performed, as previously described, in the presence or not of glucose, glutamine, and/or palmitate [17]. Wave software (version 2.6.0) was used for the data analysis.

### 2.6. Analysis of Glucose and Palmitate Uptake

Cells cultured in 96- or 24-well plates were treated, and the relative glucose or palmitate uptakes were determined, as previously described for endothelial cells using a FACS VERSE^TM^ cytometer from BD Biosciences (San Jose, CA, USA) [17]. In an additional experiment, the cells were seeded on glass cover slides and incubated with BODIPY FL C_16_, washed, fixed with 4% formalin, the nuclei were stained with Hoechst, and the covered slides were mounted (Fluoromount-G, Southern Biotech (Birmingham, AL, USA)). Photographs of the BODIPY FL C_16_ intracellular location were taken using a Leica SP8 confocal microscope (Wetzlar, Germany).

### 2.7. Lactate Production

The cells cultured in six-well plates were incubated for 30 min in the presence of different metabolic fuels. Afterwards, the media were collected and the lactate concentration was measured as previously described [17].

### 2.8. Glutamine Uptake and Oxidation

The cells cultured in 96- or 24-well plates were incubated in the presence or absence of different metabolic substrates for 30 min. The glutamine uptake and oxidation were later measured as previously described [17]. All of the assays were performed in the Radioactive Installation of the University of Málaga, authorized with reference IR/MA-13/80 (IRA-0940) for the use of non-encapsulated radionuclides.

### 2.9. RNA Isolation and Purification and cDNA Synthesis

Tri reagent and the Direct-zol™ RNA MiniPrep Kit (Zymo Research) were used according to the instructions provided by the suppliers, so as to extract the total RNA from the MDA-MB-231 cells seeded in six-well plates. Complementary DNA (cDNA) was obtained from purified RNA with the High-Capacity cDNA Reverse Transcription Kit (Applied Biosystems).

### 2.10. qPCR

The qPCR (quantitative RT-PCR) of cDNA, obtained as described above, was performed in an Eco Real-Time PCR System using KAPA SYBR Fast Master Mix (2×) Universal (KAPA Biosystems). The profile of the thermal cycling used was as follows: 95 °C for 3 min, 40 cycles of 95 °C for 10 s, and annealing temperature (Tm) for 30 s. Three independent experiments were carried out with duplicates of each sample. The obtained data were normalized using the β-actin expression as the internal control. Table 1 collects the primers sequence, Tm, and amplicon size for each gene.

### 2.11. Western Blot

Cells incubated in the presence or not of 5 mM glucose and 30 µM PD98059 for 30 min were lysed with a 2× denaturing loading buffer. The phospho-ERK and ERK (Extracellular signal-Regulated Kinases) protein levels were determined by Western blot, as previously described by us [18].

### 2.12. Statistical Analysis

The results are given as mean ± standard deviation (SD). To determine the statistical significance, one-way analysis of variance (ANOVA) or Student’s *t*-test were used, considering significant differences as those with *p* < 0.05.

## 3. Results and Discussion

### 3.1. Glucose and Glutamine Are Essential for MDA-MB-231 Cells’ Proliferation

It has been already reported that the MDA-MB-231 cells cannot grow in the absence of glucose and glutamine, getting into G2/M (G2/mitosis) cell cycle phase block after 2–4 h and inducing apoptosis [19]. Glutamine is essential for MDA-MB-231 cell growth even in the presence of glucose [20]. Nevertheless, as far as we are concerned, the proliferation of MDA-MB-231 cells has not been tested under glucose starvation in the presence of glutamine. Our results show a total dependence on both glucose and glutamine for sustaining cell growth (Figure 1A), although they were able to survive, without growing, in the presence of only glutamine in hypoxia (Figure 1B). Interestingly, glucose and glutamine were also determined to be important for the proliferation of the non-invasive, ER-positive MCF7 cell line, but these cells seemed to be more sensitive to glucose deprivation (Figure 1C). These data reinforce the fact that a distinction between different types of breast cancer cells has to be considered for the study of breast cancer progression. Furthermore, glutamine, but not glucose, withdrawal changed MDA-MB-231 cells’ morphology, making them longer and with a fusiform shape (Figure 1D). This fact may indicate changes in the cytoskeleton structure due to the inhibition of proliferation in the absence of glutamine, suggesting a more critical role of this amino acid in sustaining cell growth in these cells. Remarkably, glutamine deprivation has been seen to affect cell invasion of melanoma cells through decreasing α5 integrin expression, focal adhesion kinase (FAK) phosphorylation, and the inhibition of actin cytoskeleton remodeling [21].

### 3.2. Effect of Different Metabolic Fuels on Glucose and Glutamine Metabolism

Next, we wanted to analyze the short-time energetic metabolism of MDA-MB-231 cells. For this, we used three main metabolic fuels, namely, glucose, glutamine, and palmitate. Palmitate is known to induce apoptosis in MDA-MB-231 cells after exposures longer than 6–8 h [9]. Our data support this observation (Appendix A). For this reason, we used this FA in experiments involving exposures shorter than 2 h in a cell culture.

We first analyzed the effect on the basal oxygen consumption rate (OCR) and extracellular acidification rate (ECAR) by adding different metabolic substrates after a fasted period. Glutamine was the major oxidative substrate in MDA-MB-231 cells, as shown by the higher OCR increase after glutamine addition (*p* < 0.0001; Figure 2A). Glucose was also used as an oxidative substrate, but to a lesser extent (*p* < 0.0001; Figure 2A). Interestingly, OCR was not increased in the presence of palmitate in MDA-MB-231 cells (Figure 2A). Accordingly, highly proliferative and glycolytic cell lines are described to have a great avidity for FAs, using them for lipid biosynthesis instead of oxidation [22,23,24,25]. A combination of different metabolic substrates slightly increased the maximum OCR values (*p* < 0.05; Figure 2A). Regarding the ECAR values, the MDA-MB-231 cell line was corroborated to be highly glycolytic in the presence of glucose (*p* < 0.0001; Figure 2B), as previously described [8,26]. Glutamine also increased the ECAR values (*p* < 0.05; Figure 2B), most likely due to deprotonation of HCO_3_^−^, resulting from oxidation [27].

Additionally, we analyzed the effects of these different metabolic substrates on the uptake and production of the others. Our results showed that neither glutamine nor palmitate alone had any significant effect on glucose uptake or lactate production after 30 min (Figure 3A,B), although the combination of glucose with both glutamine and palmitate increased lactate production in these cells (*p* < 0.05; Figure 3B). Moreover, no lactate production was detected in the absence of glucose (Figure 3B). This is in contrast to the data from other tumor cell lines, such as glioblastoma cells, which are able to produce lactate from glutamine [4,28].

Regarding glutamine metabolism, we analyzed two different features, namely: glutamine uptake from the media and glutamine oxidation. Glucose did not statistically affect glutamine uptake in MDA-MB-231 cells (Figure 3C), but it decreased glutamine oxidation (*p* < 0.01; Figure 3D), indicating a preference for glycolysis versus OXPHOS from glutamine. Surprisingly, the combination of glucose and palmitate increased glutamine uptake (*p* < 0.01; Figure 3C), whereas this combination decreased glutamine oxidation (*p* < 0.001; Figure 3D). Besides the role of glutamine in feeding the tricarboxylic acid (TCA) cycle, there are other non-oxidative metabolic fates for this amino acid, such as providing nitrogen skeletons for nucleotides and glycosylation reactions, amino acid synthesis, or signal transduction, mostly involving cell proliferation [29]. Glucose metabolism is also important for anaplerosis [7]. Moreover, free FAs such as palmitate can be used for lipid synthesis, such as triacylglycerides (TAG) and phospholipids (PL). The increase in glutamine uptake in the presence of glucose and palmitate, which are both involved in biosynthetic pathways, along with a decrease in its oxidation rate could be related to an elicitation of anaplerotic metabolism.

### 3.3. Glucose Increases Palmitate Uptake in MDA-MB-231 and Other Invasive Tumor Cell Lines

We also analyzed the effect of glucose and glutamine on palmitate uptake. Glutamine did not affect palmitate uptake in MDA-MB-231 cells (Figure 4A). However, 5 mM of glucose significantly increased palmitate uptake after 30 min (*p* < 0.0001; Figure 4A). This effect on the palmitate uptake was independent of the glucose concentration (Figure 4B). As far as we are concerned, this is the first time that this effect of glucose has been documented in this cell line, although the same happens in Ehrlich ascetic tumor cells. In the last model, glucose also diminished FA oxidation (FAO), and favored the biosynthetic fate of palmitate [30]. However, the mechanism by which glucose stimulates the palmitate uptake was not determined in the Ehrlich ascetic tumor cells.

Palmitate uptake has been seen to promote invasiveness in hepatocellular carcinoma and pancreatic cancer [31,32]. For this reason, we also tested whether this effect of glucose on the palmitate uptake was found in other breast cancer cell lines, specially comparing invasive and non-invasive cell lines. For this purpose, we tested other triple-negative breast cancer cell lines, such as MDA-MB-436 and HCC1937 cells, and a non-invasive, ER-positive breast cancer cell line, MCF7. Additionally, we also performed this experiment using a mild invasive cervix adenocarcinoma cell line, HeLa, and a highly invasive neuroblastoma cell line, Kelly, along with a non-tumor cell line, such as the endothelial cell line HUVEC. Glucose increased the palmitate uptake in the two triple-negative breast cancer cell lines (*p* < 0.05), but no effect was found for the MCF7 cells. Moreover, the glucose slightly increased the palmitate uptake in the HeLa cells (*p* < 0.01) and, to a major extent, in the Kelly cells (*p* < 0.001), whereas no effect was found in the HUVECs (Figure 4C). Hence, the effect of glucose on the palmitate uptake seems to be related to the invasive capacity of the tumor cell line, independently of the type of cancer. Noticeably, intracellular palmitate has been seen to induce a pro-inflammatory response mediated through the nuclear factor kappa B (NF-κB) pathway, which could explain its importance in cancer progression [33,34].

### 3.4. The Effect of Glucose on Palmitate Uptake Is Independent of the ERK Signaling Pathway

Importantly, lipid transporters are regulated by the ERK signaling pathway. ERK phosphorylation has been found to increase CD36 expression in the membrane of muscle cells during muscle contraction [35]. However, MDA-MB-231 barely express CD36 transporters, also known as fatty acid translocase (FAT), whereas they express fatty acid binding protein 5 (FABP5) (Figure 5A) [36]. Interestingly, CD36 expression has been reported to be inversely correlated with the metastatic potential of breast cancer cell lines [37]. This fact may support the relationship between palmitate uptake, FA transporters expression, and cancer invasiveness.

The inhibition of ERK phosphorylation has been shown to diminish FABP5 expression in MCF7 cells, thus pointing to a regulation of this lipid transporter by the ERK signaling pathway [38]. Therefore, we hypothesized that this pathway could also have a role in the effect of glucose on palmitate uptake. Nevertheless, after 30 min incubation with 5 mM glucose, no effect was observed in ERK phosphorylation compared to glucose withdrawal (Figure 5B). Furthermore, we used PD98059 in order to inhibit ERK phosphorylation (Figure 5B). This inhibition did not affect the upregulation of the palmitate uptake in the presence of glucose (Figure 5C). These data indicate that glucose acts on the palmitate uptake in a way that is independent of the ERK signaling pathway. Additional experiments should be carried out in order to elucidate the exact molecular mechanism by which glucose affects palmitate uptake in these cells.

### 3.5. Glucose Is Needed for Palmitate Metabolism in MDA-MB-231 Cells

We then tested whether glucose needs to be metabolized in order to exert its effect on the palmitate uptake. The presence of the glucose analog 2-deoxyglucose (2-DG), which cannot be metabolized through glycolysis, at a concentration of 5 mM, not only failed to increase the palmitate uptake, but decreased it (*p* < 0.05; Figure 6A). Increasing the glucose concentration along with 5 mM 2-DG reestablished the increase of palmitate uptake beyond the effect of the 2-DG, totally restoring this effect when te glucose and 2-DG were present at the same concentration (Figure 6A).

Glucose is a precursor of lipid synthesis through conversion to G3P, which will incorporate two acyl-CoAs in a reaction mediated by mitochondrial glycerol-3 phosphate acyltransferase (GPAT2), generating phosphatidic acid (PA). This PA will be diverted to PL or to diacylglicerides(DAG) and triacylglicerides (TAG) synthesis [39]. Glycerol-3 phosphate dehydrogenase (GPDH) is the enzyme that converts dihydroxyacetone phosphate (DHAP) into G3P. Among its functions, this enzyme is important for the reoxidation of cytosolic NADH in glycolytic cells, as well as the regulation of cytosolic G3P, thus regulating glycolysis, lipogenesis, and OXPHOS [40]. The treatment of MDA-MB-231 cells with adipostatin A, an inhibitor of GPDH, diminished the effect of glucose on the palmitate uptake in a dose-dependent manner (*p* < 0.05; Figure 6B). This inhibition of GPDH by adipostatin would most likely lead to a lower G3P synthesis and availability, decreasing the backbone amount to which palmitoyl-CoA could bind to in order to synthetize PA in the reaction catalyzed by GPAT2. Therefore, because of the uselessness of glucose in generating G3P, it could be logical that less palmitate uptake would be necessary. This fact suggests the possible role of glucose in tumor cell proliferation and invasiveness through the modulation of lipid synthesis.

Noticeably, GPAT2 is associated with higher rates of cell proliferation and migration in cancer cells, and a higher tumorigenicity [39,41]. In accordance with our results, the GPAT2 expression was found to be much higher in MDA-MB-231 than in the HeLa and MCF7 cells lines, which present a low GPAT2 expression [39]. Indeed, glucose was seen to be incorporated into glycerol and fatty acyl chains in breast epithelial cells (HMEC), and breast tumor cell lines MCF7 and ZR75-1, but only to glycerol in MDA-MB-231 cells [42]. However, in that article, the effect of glucose on FA uptake was not studied [42]. 

Moreover, palmitate in MCF7 cells is set aside for mitochondrial oxidation. Whereas palmitate uptake has been found to be similar in MCF7 and MDA-MB-231 cell lines, only MDA-MB-231 cells, in which palmitate has a different metabolic fate, underwent apoptosis after palmitate exposure [43]. Whether glucose is involved in the pro-apoptotic effect of palmitate in MDA-MB-231 cells or if palmitate also triggers an apoptotic response in other invasive tumor cell lines needs to be further researched. 

Palmitate is known to mainly lead to PL and TAG synthesis in MDA-MB-231 cells [44]. Interestingly, in the present work, we failed to visualize the fluorescent analog of palmitate BODIPY FL C_16_ in the plasma membrane of MDA-MB-231 cells (Figure 6C), pointing to a possible accumulation of TAG instead of PL synthesis. Other authors found that MDA-MB-231 cells with an active glucose oxidative metabolism and incubated with oleate accumulated TAG inside the lipid droplets. In that work, the authors speculated that glucose was being converted to G3P in order to sustain the TAG synthesis [45]. However, this statement was not supported by any experimental data beyond the glucose oxidation rate. In the present work, we demonstrated that glucose metabolism, through the conversion of glucose to G3P, supports palmitate uptake in the MDA-MB-231 cell line, most likely followed by incorporation into G3P to form PA, the precursor of TAG.

## 4. Conclusions

In summary, the data obtained in this study point out to an essential role of glucose in the breast cancer cell line MDA-MB-231 for proliferation, glycolysis, and lipid synthesis, thus adding interesting information about the metabolic profiling of these highly proliferative and invasive, estrogen insensitive breast cancer cells [45]. However, we also show that MDA-MB-231 cells can survive up to two days in the absence of glucose, but not in the absence of glutamine. Despite the pro-apoptotic effect of palmitate after long incubation, glucose induced, in a dose-independent and ERK-independent manner, an increase in palmitate uptake after 30 min, probably for the synthesis of triglycerides. This effect of glucose was also exerted in other triple-negative breast cancer cell lines, such as MDA-MB-436 and HCC1937, and in other invasive tumor cell lines such as HeLa and Kelly, whereas no effect was found in the non-invasive breast cancer cell line MCF7. These results point to a regulation between glucose and lipid metabolism in invasive cancer cells, not given in the non-invasive tumor cells studied, and open new horizons for targeting glucose and lipid metabolism for the inhibition of cancer progression. Finally, this work warns caution in relation to many other studies of tumor metabolism that are being carried out in conditions far removed from the physiological ones.

## Figures and Tables

**Figure 1 biology-09-00016-f001:**
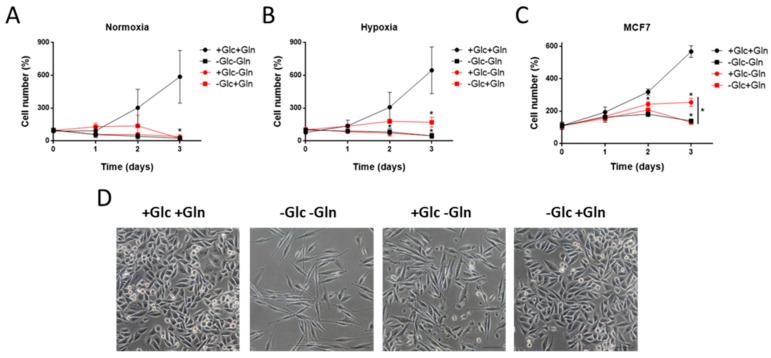
Effect of glucose and/or glutamine starvation on the proliferation of breast cancer cells. (**A**) Cell growth in normoxia and (**B**) hypoxia (1% O_2_) for MDA-MB-231 cells, and (**C**) in normoxia for MCF7 cells under different combinations of 5 mM glucose and 0.5 mM glutamine. (**D**) Representative photographs of MDA-MB-231 cells’ morphology under different glucose and/or glutamine starvation conditions for 24 h. Bar scale = 200 µm. Data are expressed as means ± standard deviation (SD) of three independent experiments. * *p* < 0.05 versus glucose and glutamine conditions.

**Figure 2 biology-09-00016-f002:**
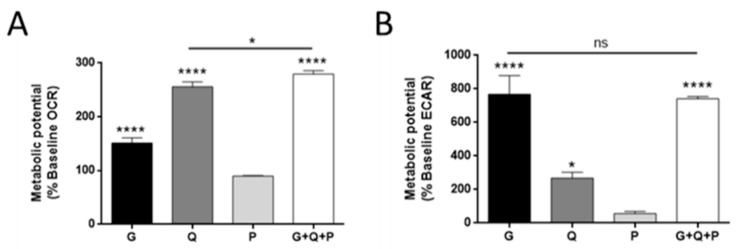
Energetic metabolism in MDA-MB-231 cells. (**A**) Oxygen consumption rate (OCR) and (**B**) extracellular acidification rate (ECAR). Three initial measurements were made in MDA-MB-231 cells incubated in media without glucose, glutamine, and palmitate. These measurements were considered the baseline. Then, 5 mM glucose, 0.5 mM glutamine, and/or 0.5 mM palmitate were injected and additional measurements were taken. Data are expressed as means ± SD of three independent experiments with triplicate samples each. * *p* < 0.05, **** *p* < 0.0001 versus control without any metabolic substrate. G—glucose; Q—glutamine; P—palmitate.

**Figure 3 biology-09-00016-f003:**
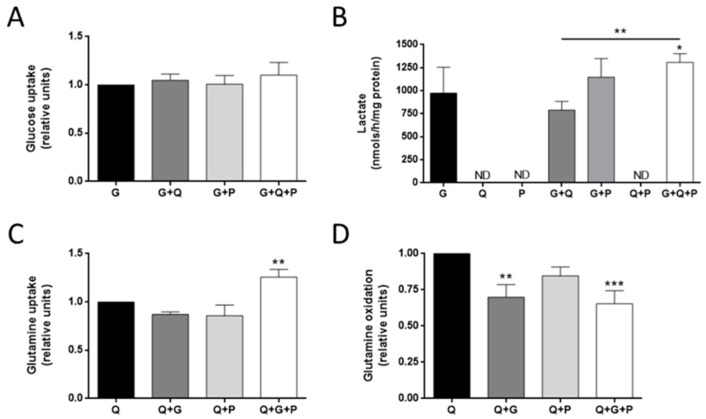
Effect of different metabolic fuels on glucose and glutamine metabolism in MDA-MB-231 cells. (**A**) Glucose uptake, (**B**) lactate secretion, (**C**) glutamine uptake, and (**D**) glutamine oxidation were determined after 30 min fast and 30 min incubation in the presence of 5 mM glucose, 0.5 mM glutamine, and/or 0.5 mM palmitate. Data are expressed as means ± SD of three independent experiments. * *p* < 0.05, ** *p* < 0.01, *** *p* < 0.001 versus glucose (**A**,**B**) or glutamine (**C**,**D**) condition. G—glucose; Q—glutamine; P—palmitate.

**Figure 4 biology-09-00016-f004:**
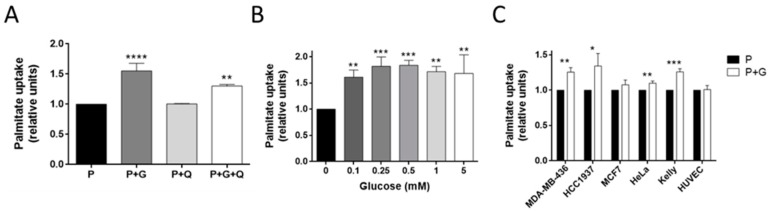
Effect of glucose on palmitate uptake. (**A**) Palmitate uptake in the presence of 5 mM glucose and/or 0.5 mM glutamine, or (**B**) with different concentrations of glucose in MDA-MB-231 cells. (**C**) Palmitate uptake in the presence or not of 5 mM glucose in MDA-MB-436, HCC1937, MCF7, HeLa, Kelly, and human umbilical vein endothelial cells (HUVEC) cell lines. All of the experiments were performed after 30 min of fasting and 30 min of incubation with 0.5 mM palmitate, 2 μM BODIPY (BOron-DIPYrromethene) FL C_16_, and indicated concentrations of glucose. Data are expressed as means ± SD of three independent experiments. * *p* < 0.05, ** *p* < 0.01, *** *p* < 0.001, **** *p* < 0.0001 versus palmitate condition. G—glucose; Q—glutamine; P—palmitate.

**Figure 5 biology-09-00016-f005:**
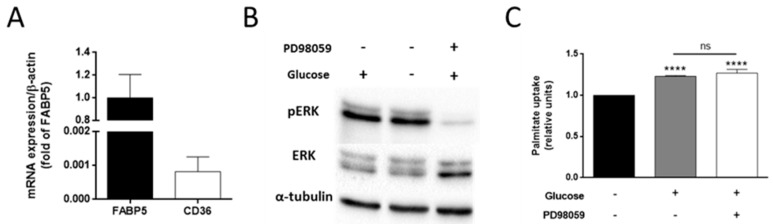
Effect of glucose on the ERK (Extracellular signal-Regulated Kinases) signaling pathway in MDA-MB-231 cells. (**A**) mRNA expression of FABP5 and CD36 transporters. (**B**) ERK phosphorylation and (**C**) palmitate uptake in the presence of 5 mM glucose alone or along with 30 μM of PD98059. (**B**,**C**) were performed after 30 min of fasting and 30 min of incubation with 0.5 mM palmitate and 2 μM BODIPY (BOron-DIPYrromethene) FL C_16_. Data are expressed as means ± SD of three independent experiments. **** *p* < 0.0001 versus condition without glucose and PD98059.

**Figure 6 biology-09-00016-f006:**
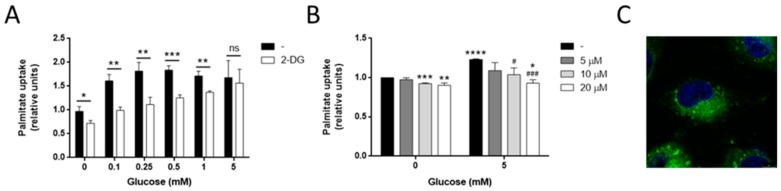
Utilization of glucose for palmitate metabolism in MDA-MB-231 cells. (**A**) Palmitate uptake in the presence of different concentrations of glucose along with 5 mM 2-deoxyglucose (2-DG) or (**B**) different concentrations of adipostatin A. (**C**) Representative photograph of BODIPY FL C_16_ intracellular location in MDA-MB-231 cells. Bar scale = 8 μm. All of the experiments were performed after 30 min of fasting and 30 min of incubation with 0.5 mM palmitate, 2 μM BODIPY FL C_16_, and indicated concentrations of glucose. Data are expressed as means ± SD of three independent experiments. * *p* < 0.05, ** *p* < 0.01, *** *p* < 0.001, **** *p* < 0.0001 versus condition without 2-DG (**A**) or with only palmitate (**B**). # *p* < 0.05, ### *p* < 0.001 versus palmitate and glucose (**B**).

**Table 1 biology-09-00016-t001:** Primers used for qPCR.

Gene	Primers	Annealing Temperature (°C)	Amplicon Size (bp)
β-actin	Fw: GACGACATGGAGAAAATCTG Rv: ATGATCTGGGTCATCTTCTC	60	131
FABP5	Fw: AAGATGGGAAATTAGTGGTG Rv: AACAGTATGGAGATTTGCTC	60	153
CD36	Fw: AGCTTTCCAATGATTAGACG Rv: GTTTCTACAAGCTCTGGTTC	60	111

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
