# Peer review of "Glucose Favors Lipid Anabolic Metabolism in the Invasive Breast Cancer Cell Line MDA-MB-231"

_biology, 2020, doi:10.3390/biology9010016_

Round 1
Reviewer 1 Report
The manuscript is interesting and novedously. In general the structure and redaction is good. The title is appropriate with the principal results. The introduccion is good. The methodology is sufficient. The results are interesting and supported the discussion. Also, the authors studied relevant aspects related with the energy cell metabolism in cancer. However, I have the nexts comments:
Major comments:
1. The authors need improved the discussion, especially in the aspects related with the molecular mechanism involved in the responses.
1.1. The changes in the expression of FABPS and CD36 are directly related to the modification in the activity of PPAR-gamma transcription factor. This point is importante, because the excess of energy (such as fat) induced the reduction in the activity of this transcription factor in the cell.
1.2. The increment in palmitic acid (C16:0) induced the alteration in the energy cell metabolism. To respect, the increment of palmitic acid up-regulate the inflammatory response through the activation of NF-kB. Also, this change is directly related with the reducction in the content of n-3 fatty acids. Finally, this pro-inflamatory response in the cell can by modulated by the specific lipid mediators fron n-3 fatti acids (resolvins and protectins).
References sugested:
Echeverría et al., Reduction of high-fat diet-induced liver proinflammatory state by eicosapentaenoic acid plus hydroxytyrosol supplementation: involvement of resolvins RvE1/2 and RvD1/2. J Nutr Biochem. 2019;63:35-43.
Minor comments.
1. Improve the redaction of the objective to the study.
2. Check the English redaction, and use of abbreviations in all manuscript.
Author Response
Reply to Reviewer 1
Comments and Suggestions for Authors
The manuscript is interesting and novedously. In general the structure and redaction is good. The title is appropriate with the principal results. The introduccion is good. The methodology is sufficient. The results are interesting and supported the discussion. Also, the authors studied relevant aspects related with the energy cell metabolism in cancer. However, I have the nexts comments:
Major comments:
The authors need improved the discussion, especially in the aspects related with the molecular mechanism involved in the responses.1.1. The changes in the expression of FABPS and CD36 are directly related to the modification in the activity of PPAR-gamma transcription factor. This point is importante, because the excess of energy (such as fat) induced the reduction in the activity of this transcription factor in the cell.
We really appreciate Reviewer 1 comments. However, we consider that mentioning PPAR-gamma in our manuscript without providing with any experimental result could be considered as excessive speculation. In the time given for answering to this revision (10 days) we could not acquire the required materials in order to study the role of PPAR-gamma in the induction of palmitate uptake mediated by glucose. Nevertheless, we consider that this is a very interesting point and we thank Reviewer 1 for this suggestion.
1.2. The increment in palmitic acid (C16:0) induced the alteration in the energy cell metabolism. To respect, the increment of palmitic acid up-regulate the inflammatory response through the activation of NF-kB. Also, this change is directly related with the reducction in the content of n-3 fatty acids. Finally, this pro-inflamatory response in the cell can by modulated by the specific lipid mediators fron n-3 fatti acids (resolvins and protectins).
We have modified the discussion of the amended manuscript including the relation between palmitic acid and inflammation.
References sugested:
Echeverría et al., Reduction of high-fat diet-induced liver proinflammatory state by eicosapentaenoic acid plus hydroxytyrosol supplementation: involvement of resolvins RvE1/2 and RvD1/2. J Nutr Biochem. 2019;63:35-43.
We have cited it and included it in the list of references.
Minor comments:
Improve the redaction of the objective to the study.Redaction of the objective has been modified in the amended version of the manuscript as follows: “In this work, we wanted to study two features of MDA-MB-231 cells metabolism: 1) the effect of glucose and/or glutamine starvation on proliferation, and 2) the possible alterations in metabolism after short-time incubation with different combinations of glucose, glutamine and/or palmitate. In order to avoid possible interferences with other metabolites from culture media and serum, in this work we performed the short-time experiments in very restrictive conditions, although it should be taken into account that other metabolites may also be present in an individual. Thus, this kind of methodological approximation allows the determination of the short-time preference of metabolic substrate in these cells when only these fuels are present at physiological concentration after a fast period [12,13]. Combining long and short-term experiments covers different possible scenarios, such as the short-term response of the cells to a drastic nutritional change (their immediate fuel preference) and the long-term consequence of that change (their ability to adapt to a different nutritional condition). Increasing knowledge in the metabolic flexibility of cancer cells could allow the design of specific therapies in order to inhibit tumor progression. Therefore, our study may lead to further research in the metabolomics of breast cancer cells and the inhibition of breast cancer progression.”
Check the English redaction, and use of abbreviations in all manuscript.The text has been corrected according to this comment.
We thank Reviewer 1 for his/her comments. We think that they have been really helpful to improve our work.
Reviewer 2 Report
Ocaña C et al.,
The authors studied the short-time preference of metabolic substrate in MDA-MB-231 cell line. Results point out to a preference for glucose but also a strong dependence on both glucose and glutamine to proliferate. Glucose increases palmitate uptake in MDA-MB-231 and other invasive cancer cell lines. Neither glutamine nor palmitate alone had any significant effects on glucose uptake or lactate production. Glucose acts on palmitate uptake in a way that is independent of the ERK signaling and CD36 activity.
Comments:
Figure 5. The MDA-MB-231 cell line was reported expressing CD36 and thus could be equipped to introduce FFA (DOI: 10.1038 / s41389-018-0107-x). Please check again CD36 expression or introduce a specific comment..
What are the possible benefits resulting from the study? Please discuss this point.
Some concepts have been explained in a user-friendly format. As an example the sentence: This allow us to determine the short-time preference ….(Rows 54-56).
Author Response
Reply to Reviewer 2
Comments and Suggestions for Authors
Ocaña C et al.,
The authors studied the short-time preference of metabolic substrate in MDA-MB-231 cell line. Results point out to a preference for glucose but also a strong dependence on both glucose and glutamine to proliferate. Glucose increases palmitate uptake in MDA-MB-231 and other invasive cancer cell lines. Neither glutamine nor palmitate alone had any significant effects on glucose uptake or lactate production. Glucose acts on palmitate uptake in a way that is independent of the ERK signaling and CD36 activity.
Comments:
Figure 5. The MDA-MB-231 cell line was reported expressing CD36 and thus could be equipped to introduce FFA (DOI: 10.1038 / s41389-018-0107-x). Please check again CD36 expression or introduce a specific comment.
We really appreciate this comment from Reviewer 2. However, CD36 expression reported in that paper is quite low. Others were not able to detect CD36 in MDA-MB-231 cells (Guaita-Esteruelas et al 2017 Mol. Carcinog. 56; Uray et al 2004 Cancer Lett. 207). Please find below the amplification curves for the experiment of mRNA expression of FABP5 and CD36 by qPCR in these cells. As you can see, some amplification was detected for CD36. Nevertheless, Cq for this gene was greater than 35. We usually do not consider this kind of data as reliable due to the high sensibility of this technique. We have now modified Figure 5A in the amended manuscript and considered those high Cq for CD36. We now represent those data as fold of FABP5 so that it can be seen how CD36 is almost negligible in comparison with FABP5 expression. Therefore, it seems unlikely that CD36 helps fatty acid uptake in these cells in a significant manner. The discussion of the manuscript has also been modified accordingly.
Figure. β-actin (blue), CD36 (green) and FABP5 (red) mRNA amplification blots in MDA-MB-231 cells.
Figure (corresponding to new Figure 5A of the amended manuscript). mRNA expression of FABP5 and CD36 in MDA-MB-231 cells.
What are the possible benefits resulting from the study? Please discuss this point.
We consider that performing these short-term experiments allows the study of the immediate effect of a drastic increase/decrease in a metabolite concentration (such as glucose or glutamine) on different metabolic features. Combining long and short-term experiments covers different possible scenarios, such as the short-term response of the cells to a drastic nutritional change (their immediate fuel preference) and the long-term consequence of that change (their ability to adapt to a different nutritional condition). Increasing knowledge in the metabolic flexibility of cancer cells could allow the design of specific therapies in order to inhibit tumor progression. We believe that our study may lead to further research in the metabolomics of breast cancer cells and the inhibition of breast cancer progression.
Some concepts have been explained in a user-friendly format. As an example the sentence: This allow us to determine the short-time preference ….(Rows 54-56).
We have revised the text and changed this kind of user-friendly sentences in the amended version of the manuscript.
We thank Reviewer 2 for his/her comments. We think that they have been really helpful to improve our work.
Reviewer 3 Report
Major comments:
The authors provided data regarding glucose, glutamine, and palmitate uptake in MDA-MB-231 breast cancer cell line. However, it is quite difficult to follow how such metabolic profile of a single cell line can be translated to understanding the metabolic reprogramming in TNBC tumors and tumor microenvironment. Also, wondering how results of this study can contribute to answering the role of tumor metabolome in TNBC invasiveness and the disease progression.
Minor comments are listed below:
Is this cell line MDA-MB-231 cell line used in this study representing the metabolomes of TNBC? If so, did you confirm the consistency in metabolic profiles in other TNBC cell lines? If not, how do we interpret your data to understand metabolic reprogramming of TNBC or a certain breast cancer subtype?What is the point of a single cell line analysis for understanding the role of metabolomic changes in breast cancer aggressiveness? Please elaborate the study by incorporating other BREAST cancer cell lines (invasive vs. non-invasive, or TNBC or ER+) in this study.
Lines 226-227: What is the rationale behind analyzing other cancer types (e.g. Kelly etc.)? How is this useful to understand breast cancer aggressiveness? How palmitate uptake is related to invasive phenotypes? Please elaborate data interpretation.
Glutamine dependency is MDA-MB-231 specific? What about it in MCF-7 cell line?
Lines 146-148: Provide a clarity of this statement. How glutamine depletion affects cell morphology or cytoskeletal structure? Elaborate data interpretation.
Lines 151-154: What is the point of analyzing the metabolomic changes under a short exposure of palmitate? How this experimental condition can be pathophysiologically relevant to breast cancer metabolomes, particularly in the context of disease progression?
Line 160: Where are the statistical values in the figure?
Figure 1C: The cell pics of +Glc+Gln and -Glc+Gln showed the similar cell numbers. Do you think these pictures match with the results of Figure 1A and Figure 1B? Revise them.
Lines 184-185: Glucose is missing in the sentence. It should go, “Although combination of glucose with both glutamine and palmitate increased…”
Lines 186-187: Specify other tumor cell lines.
Sentence redundancy: Lines 229-232 and Lines 277-282. Rephrase them.
Author Response
Reply to Reviewer 3
Comments and Suggestions for Authors
Major comments:
The authors provided data regarding glucose, glutamine, and palmitate uptake in MDA-MB-231 breast cancer cell line. However, it is quite difficult to follow how such metabolic profile of a single cell line can be translated to understanding the metabolic reprogramming in TNBC tumors and tumor microenvironment. Also, wondering how results of this study can contribute to answering the role of tumor metabolome in TNBC invasiveness and the disease progression.
We chose the MDA-MB-231 cell line for being one of the most typical models for highly-invasive breast cancer cells. Most of the results showed in our manuscript were obtained using these cells. However, we agree with Reviewer 3 that the use of only one cell line cannot lead to a strong conclusion. We really appreciate his/her comment and now we have included some results from additional breast cancer cell lines, as suggested by Reviewer 3 in this and other comments (see below).
Minor comments are listed below:
Is this cell line MDA-MB-231 cell line used in this study representing the metabolomes of TNBC? If so, did you confirm the consistency in metabolic profiles in other TNBC cell lines? If not, how do we interpret your data to understand metabolic reprogramming of TNBC or a certain breast cancer subtype?
What is the point of a single cell line analysis for understanding the role of metabolomic changes in breast cancer aggressiveness? Please elaborate the study by incorporating other BREAST cancer cell lines (invasive vs. non-invasive, or TNBC or ER+) in this study.
Following Reviewer 3 suggestion, we have included some results from additional breast cancer cell lines. We studied the effect of glucose on palmitate uptake (the main result of the manuscript) in other two triple-negative breast cancer cell lines. These cells lines are MDA-MB-436 and HCC1937. For both cell lines, we found out that glucose significantly increased palmitate uptake. These results coincide with the ones from MDA-MB-231 cells. Please find the new results below. On the other hand, we already had results from the non-invasive, ER+ breast cancer cell line MCF7. Results from these cells indicated that glucose does not significantly increase palmitate uptake. We have included the new results as part of the new Figure 4C of the amended version of the manuscript.
Figure. Palmitate uptake in the absence (P) or presence (P+G) of 5 mM glucose in MDA-MB-436 and HCC1937 breast cancer cell lines. *p < 0.05 and **p < 0.01 compared to condition without glucose.
Lines 226-227: What is the rationale behind analyzing other cancer types (e.g. Kelly etc.)? How is this useful to understand breast cancer aggressiveness? How palmitate uptake is related to invasive phenotypes? Please elaborate data interpretation.
For this work, we included different types of cancer cell types with different grades of invasiveness. That way, we could try to stablish a correlation between palmitate uptake and tumor invasion. For example, expression of the fatty acid transporter CD36 has been reported to be inversely correlated with the metastatic potential of breast cancer cell lines (Uray et al 2004 Cancer Lett. 207). On the other hand, palmitate uptake has been seen to promote invasiveness in hepatocellular carcinoma and pancreatic cancer (Binker-Cosen et al 2017 Biochem. Biophys. Res. Commun. 484; Nath et al 2015 Sci. Rep. 5). Accordingly, CD36 expression in MDA-MB-231 cells is almost negligible, and our results show that glucose increased palmitate uptake in this highly-invasive tumor cell line. We have followed Reviewer 3 suggestion and now data from other breast cancer cell lines are included in the amended version of this manuscript, supporting the role of palmitate uptake in the invasiveness of breast cancer. We have now added these aspects and the new data to the Discussion of the amended version of the manuscript.
Glutamine dependency is MDA-MB-231 specific? What about it in MCF-7 cell line?
We already tested glutamine dependency for proliferation in other tumor and non-tumor cell lines. We previously showed that glutamine is essential for endothelial cells and the human cervix adenocarcinoma cell line HeLa (Ocaña et al 2019 Biomolecules 9). Moreover, in the Results and Discussion section of the manuscript we referenced an article where the authors found out that glutamine was essential for MDA-MB-231 cells proliferation even when glucose was available (Korangath et al 2015 Clin. Cancer Res. 21). We now have included the results obtained for the dependence of MCF7 on glucose and/or glutamine for proliferation. Please find these results below. Surprisingly, in this tumor cell model glutamine is also needed for cell proliferation, but these cells seem to be more sensitive to glucose deprivation. These data reinforce the fact that a distinction between different types of breast cancer cells has to be considered. These results have been included in the amended version of the manuscript.
Figure (corresponding to new Figure 1C of the amended manuscript).
Lines 146-148: Provide a clarity of this statement. How glutamine depletion affects cell morphology or cytoskeletal structure? Elaborate data interpretation.
For this work, we did not studied the effect of glutamine depletion on cytoskeletal structure. However, there is an evident change of cell shape under these conditions in MDA-MB-231 cells (Figure 1D of the amended manuscript). Another work reported that glutamine restriction diminished α5 integrin expression, FAK phosphorylation and inhibited actin cytoskeleton remodeling in melanoma cells (Fu et al 2004 Clin. Exp. Metastasis 21). We have now included this in the discussion of the amended manuscript.
Lines 151-154: What is the point of analyzing the metabolomic changes under a short exposure of palmitate? How this experimental condition can be pathophysiologically relevant to breast cancer metabolomes, particularly in the context of disease progression?
For this work, we used palmitate for a short time due to the reported pro-apoptotic effect of this fatty acid. However, it is known that other fatty acids, such as oleate, rescue this pro-apoptotic effect (Hardy et al 2003 J. Biol. Chem. 278). In an individual, not only palmitate would be present in the blood and extracellular media, but also other fatty acids, so that this pro-apoptotic effect is not likely exerted. On the other hand, we consider that performing these short-term experiments allows the study of the immediate effect of a drastic increase/decrease in a metabolite concentration (such as glucose or glutamine) on different metabolic features. Combining long and short-term experiments covers different possible scenarios, such as the short-term response of the cells to a drastic nutritional change (their immediate fuel preference) and the long-term consequence of that change (their ability to adapt to a different nutritional condition). Increasing knowledge in the metabolic flexibility of cancer cells could allow the design of specific therapies in order to inhibit tumor progression. We believe that our study may lead to further research in the metabolomics of breast cancer cells and the inhibition of breast cancer progression.
Line 160: Where are the statistical values in the figure?
We are sorry for this mistake. Figure 1 has been modified in the amended version of the manuscript including the statistical values.
Figure 1C: The cell pics of +Glc+Gln and -Glc+Gln showed the similar cell numbers. Do you think these pictures match with the results of Figure 1A and Figure 1B? Revise them.
We are sorry for this confusing figure. Those pictures are from cells cultured under different glucose/glutamine conditions for just 24 h. At that point, cells in the absence of glutamine (-Glc-Gln and +Glc-Gln) completely stopped growing and showed a different cell shape. On the other hand, in the presence of glutamine (+Glc+Gln and -Glc+Gln) cell number was similar between those two conditions. At 48 h, cells grown in the absence of both glucose and glutamine started to die, whereas cell number of cells grown with glucose and glutamine kept increasing. Please find the pictures at 48 below and compare them with the cell number data at day 2. We have now modified the figure legend of Figure 1 in the amended version of the manuscript indicating when those pictures were taken.
Figure. Representative pictures of MDA-MB-231 cells under different nutritional conditions for 48 h.
Lines 184-185: Glucose is missing in the sentence. It should go, “Although combination of glucose with both glutamine and palmitate increased…”
We have corrected this sentence in the amended manuscript.
Lines 186-187: Specify other tumor cell lines.
We have modified this sentence in the amended manuscript.
Sentence redundancy: Lines 229-232 and Lines 277-282. Rephrase them.
We are sorry for this redundancy. We have now corrected the discussion avoiding duplicated information in the amended version of the manuscript.
We thank Reviewer 3 for his/her carefully revision of this manuscript. We think that it has been really helpful to improve our work.
Round 2
Reviewer 3 Report
The authors have addressed concerns in a proper manner.